# The efficacy of Topical Clascoterone versus systematic spironolactone for treatment of acne vulgaris: A systematic review and network meta-analysis

**Mohammad A. Basendwh** [1] \*, **Amer Abdulaziz Alharbi**[2], **Sarah Abdulla Bukhamsin**[3], **Rahaf Abdulrashid Abdulwahab**[4], **Sarah Abdullah Alaboud**[4]

**1** Dermatology Department, King Fahad Armed Forces Hospital, Jeddah, Saudi Arabia, **2** Dermatology Department, King Saud University Medical City, Riyadh, Saudi Arabia, **3** Family Medicine Department, Al-Ahsaa Health Cluster, Al-Ahsaa, Saudi Arabia, **4** College of Medicine, Umm Alqura University, Makkah, Saudi Arabia

\* drmab1@hotmail.com, kfafh_derma@kfafh.med.sa

**Data Availability Statement:** All relevant data are on OSF: https://osf.io/jb9ra/.

## Abstract

### Objectives

This study aimed to evaluate the effectiveness of topical clascoterone (TC) compared to oral spironolactone for acne vulgaris treatment.

### Methods

A computerized search through PubMed/MEDLINE, SCOPUS, and the Cochrane Library was conducted to find relevant papers. We used the "netmeta" and "meta" packages for network meta-analysis (NMA) in RStudio 1.2.5019 (2009–2019 RStudio, Inc.) to conduct all of our statistical tests.

### Results

Seven articles (n = 2,006 patients) were included. The fixed-effect size showed that TC 1% bis in die (BID) showed potential effectiveness in reducing the inflammatory and non-inflammatory lesion count compared to placebo (Standardized mean difference, SMD = -0.27, 95% CI: -0.36 to -0.17) and (SMD = -0.31, 95% CI: -0.41 to -0.22), respectively. The random-effect size showed that TC 1% BID was significantly associated with a 12-week treatment success compared to placebo (Odds ratio, OR = 2.44, 95% CI: 1.12 to 5.30). Spironolactone 200 mg was associated with a significant reduction in total lesion count (SMD = -4.46, 95% CI: -5.60 to -3.32).

### Conclusion

TC appears to reduce both inflammatory and non-inflammatory lesion count and may lead to treatment success. Spironolactone at 200 mg showed potential effectiveness in terms of

**Funding:** The authors received no specific funding for this work.

**Competing interests:** The authors have declared that no competing interests exist.

total lesion count reduction. These results suggest that both TC and Spironolactone could be beneficial in treating patients with acne vulgaris.

## 1. Introduction

Acne Vulgaris is a chronic inflammatory skin disorder developing in the pilosebaceous unit characterized by increased sebum production and bacterial colonization [1, 2]. In young adults and adolescents, it is one of the most frequent skin diseases [3]. There are over 50 million new instances of acne each year in the United States [4]. The estimated global prevalence of acne vulgaris is 9.4% [5]. From 1990 to 2019, the global number of incident acne vulgaris cases increased from 79.7 (95% uncertainty interval (UI), 67.0–91.1) to 117.4 (95% UI, 103.0–133.7) million cases [6]. Scarring and hyperpigmentation caused by acne may have psychological consequences. Further, compared to those without acne, patients who suffered from acne were more likely to experience low self-esteem, depression, and anxiety [7].

Both male and female acne pathogenesis are largely influenced by endogenous androgens [8]. Hair follicles become blocked due to sebum overproduction, hyperkeratinization, and androgen-induced inflammation, creating an environment favorable to the colonization and infection of Cutibacterium acnes [9]. Cysts, nodules, papules, pustules, and acne comedones may all arise as a result of the aforementioned acnegenic processes. Acne may be treated topically for mild cases or with a combination of topical and systemic medications for cases with moderate to severe severity, as recommended by current treatment recommendations [10].

Systematic therapy, including oral antibiotics such as tetracycline and clindamycin, is used in acne vulgaris therapy. Additionally, oral androgen receptor blockers such as oral contraceptives, spironolactone, and antiandrogen effectively treat acne. The mechanism of action of Spironolactone refers to the blockage of androgen receptors and inhibition of aldosterone production. Several recommendations advocate spironolactone as an alternative to antibiotics for females with moderate to severe acne [10, 11]. However, there is no consensus on the effectiveness of Spironolactone for treating female acne due to the lack of well-designed trials [12].

Clascoterone is a novel treatment that acts as an androgen receptor inhibitor, preventing dihydrotestosterone (DHT) from binding to androgen receptors in the skin and thereby lowering DHT's proinflammatory and sebum-inducing actions inside the pilosebaceous unit [13–16]. Skin and plasma esterases quickly hydrolyze clascoterone to cortexolone, an inactive metabolite present in all human cells and organs [16], making it associated with less systemic side effects, unlike oral antiandrogens [9]. Clascoterone works at the site of administration with low systemic exposure. No significant clinical side effects, such as testosterone fluctuations or extended hypothalamic-pituitary-adrenal axis activation, have been recorded [13–15]. Multiple cellular and molecular pathways may contribute to the localized reduction of acne lesions after topical administration of clascoterone cream. Clascoterone, for example, decreased sebum production and inflammatory cytokines in cultured primary human sebocytes [14]. Upon our search, there is no network meta-analysis (NMA) comparing topical clascoterone (TC) and oral spironolactone. This study aimed to evaluate the effectiveness of TC compared to oral spironolactone for acne vulgaris treatment.

## 2. Methods

We reported this study in accordance with the "Preferred Reporting Items for Systematic Reviews and Meta-Analyses (PRISMA) extension statement for Network Meta-analyses of

Health Care Interventions" [17]. Also, we used the "Cochrane Handbook for Systematic Reviews of Interventions" while implementing this study [18].

## 2.1. Search strategy

Up to June 2022, we searched through PubMed/MEDLINE, SCOPUS, and the Cochrane Library to find relevant papers. For our analysis, we utilized the following search terms: Randomized clinical trial (RCT), clascoterone, spironolactone, and acne.

## 2.2. Eligibility criteria

The included RCTs met specific criteria: they involved patients with acne vulgaris; utilized interventions such as TC at doses of 0.05% bis in die (BID), 0.1% BID, 1% BID, and 1% once daily (ODS), and/or oral Spironolactone at 25 mg, 50 mg, and 200 mg, with a comparison to placebo; and focused on outcomes like total acne lesion count (TLC), inflammatory lesion count (ILC), non-inflammatory lesion count (NILC), and Investigator's Global Assessment (IGA) of acne severity. Animal studies, in vivo studies, non-RCTs, conference abstracts, and studies not in English were excluded.

## 3. Study selection

Initially, we eliminated duplicates and then subjected all citations to a two-step (title/abstract and full-text) screening process. The selection of studies was carried out by two independent reviewers, and any disputes were discussed and resolved by a senior reviewer.

## 4. Risk of bias assessment

We used the Cochrane risk of bias (ROB-II) assessment tool to assess the risk of bias in the included studies. ROB-II includes the following domains: selective outcome reporting, incomplete outcome data, blinding of outcome assessment, blinding of participants and personnel, allocation sequence concealment, sequence generation, and other potential sources of bias. The authors' judgment could be regarded as 'Low risk,' 'High risk,' or 'Unclear risk of bias in selected bias domains.

## 5. Data extraction

To collect the following types of information, we utilized an offline Excel sheet: Study characteristics, including study ID, major findings, sample size, intervention groups, year of publication, and country of origin; Participant characteristics, including mean age and race; Types of intervention and comparators, including TC, Spironolactone, and placebo; and Outcome measures: the efficacy of interventions in reducing the TLC, ILC, and NILC, as well as the success rate based on the IGA.

## 6. Statistical analysis

In this NMA, we derived summary measures using the Standardized Mean Difference (SMD). If variance data wasn't available as a standard deviation, algebraic formulas or other approximate methods were employed to obtain a suitable value. We used a fixed-effects model by default to calculate weighted SMDs and 95% Confidence Intervals (CI) in NMA. In cases of heterogeneity among the included studies ($I^2$ is >50% and p-value is <0.10), a random-effects model was employed. The $I^2$ and $Tau^2$ statistics were utilized to express the inter-trial heterogeneity. To synthesize the current evidence, we calculated both direct and indirect effect models. Any discrepancy between these two estimates was considered as a measure of

inconsistency and was identified using a global inconsistency test with a fitted design-by-treatment model. We ranked different therapies using the frequentist "netrank" and "rankogram" functions, which map interventions according to their relative effect. Here, a higher P-score corresponds to a superior intervention. Concerning IGA, a random-effects model was used to compute the Odds Ratio (OR) of achieving a success rate. To separate direct and indirect evidence further in NMA, we used the "netsplit" tool. We carried out all statistical tests in RStudio (version 1.2.5019) using the "netmeta" and "meta" packages, specifically designed for NMA.

# 3. Result

## 3.1. Study selection

Based on our literature search, we found a total of 261 relevant citations. After removing duplication, 210 articles underwent title/abstract screening. Then, 187 studies were deemed ineligible to our criteria. The full-text screening was performed on 23 articles, and 14 studies were excluded. Finally, seven articles (n = 2,006 patients) were included in the qualitative (systematic review) and five articles in the quantitative synthesis (meta-analysis). *Fig 1* shows the PRISMA flow diagram of included studies.

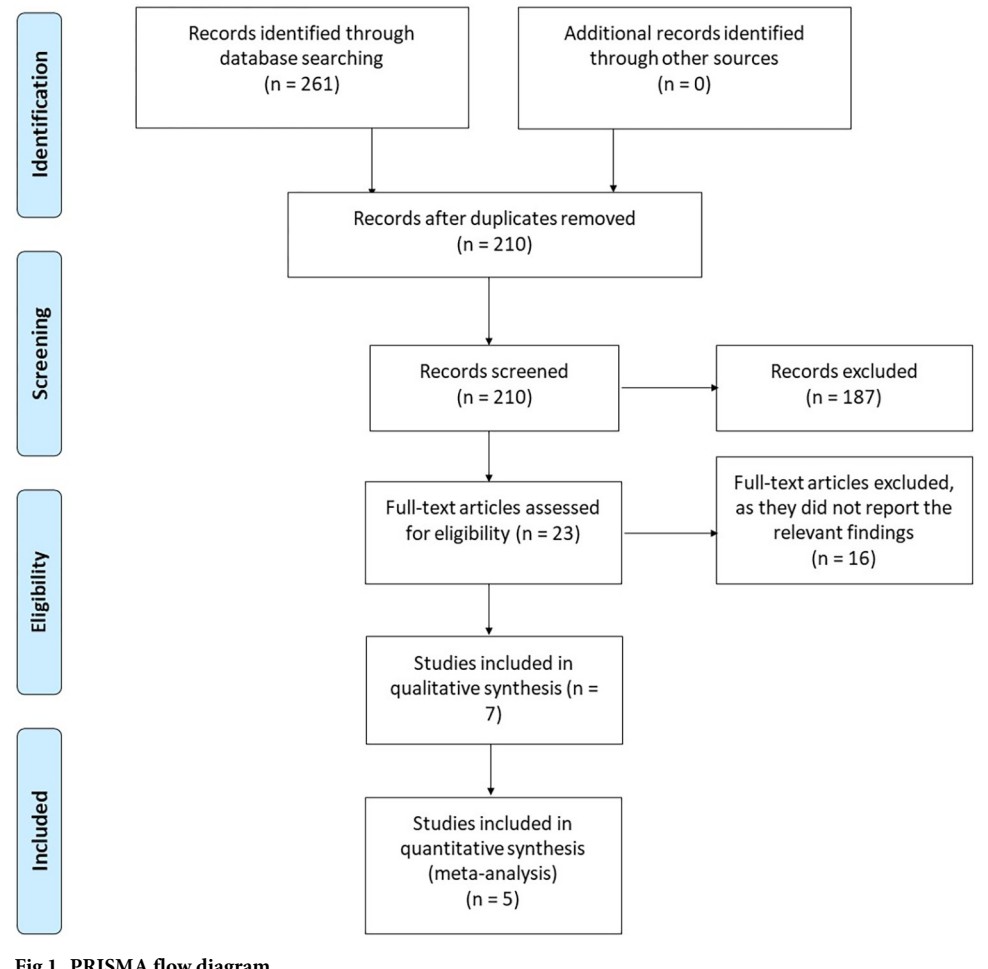

**Fig 1. PRISMA flow diagram.**

### 3.2. Characteristics of included studies and patients

Overall, the age of participants ranged from 9–45 years, with moderate to severe acne scores. All studies were conducted for 12 weeks except for Trifu *et al.* 2011 [19], Goodfellow *et al.* 1984 [20], and Muhlemman *et al.* 1986 [21], which were conducted for eight weeks, three months, and six months respectively. It is worth noting that the baseline characteristics of all patients across the included studies did not show that any of the used regimens are contraindicated in the other group. *Table 1* summarizes the characteristics of included studies and patients.

### 3.3. Quality of the included studies

Based on the ROB-II tool, there was no detected risk of selection bias, performance bias, detection bias, or attrition bias. On the other hand, we found a high risk of reporting bias in Goodfellow et al. and Patiyasikunt et al. *Fig 2* shows the domains of the ROB-II tool.

### 3.4. NILC

*Fig 3A* presents the network graph of analyzed regimens. The fixed-effect size showed that TC 1% BID has potential effectiveness in reducing the NILC compared to placebo (SMD = -0.31, 95% CI: -0.41 to -0.22), *Fig 4A*. Other regimens showed non-significant efficacy compared to placebo. The analysis showed no evidence of heterogeneity (Tau$^2$ = 0%, I$^2$ = 0%, 95% CI: 0% to 64.8%) and consistent (Q = 4.57, p = 0.80). The ranking analysis demonstrated that TC 1% BID was the best regimen (P-score = 1.00), followed by TC 0.1% BID (P-score = 0.55), TC 1% ODS (P-score = 0.44), placebo (P-score = 0.32), and TC 0.5% BID (P-score = 0.19). The net split analysis highlighted that TC 1% BID was associated with reduced NILC compared to TC 1% ODS (SMD = -0.28, 95% CI: -0.47 to -0.10), *S1 Fig in S1 File*. Moreover, compared with TC 1% BID, both TC 0.1% BID and TC 0.5% BID were associated with increased NILC (SMD = 0.25, 95% CI: 0.07 to 0.44) and (SMD = 0.35, 95% CI: 0.17 to 0.54), respectively (*Table 2*).

### 3.5. TLC

Compared to placebo, the effect estimates showed that Spironolactone 200 mg was associated with a significant reduction in TLC (SMD = -4.46, 95% CI: -5.60 to -3.32). Other regimens showed non-significant efficacy compared to placebo, *Fig 4B*. The analysis showed no evidence of heterogeneity (Tau$^2$ = 0%, I$^2$ = 0%, 95% CI: 0% to 89.6%) and consistent (Q = 0.01, p = 0.99). The network graph is presented in *Fig 3B*. The ranking analysis demonstrated that Spironolactone 200 mg was associated with highest ranking (P-score = 1.00), followed by TC 1% BID (P-score = 0.70), Tretinoin 0.05% (P-score = 0.54), Spironolactone 25 mg (P-score = 0.26), and Spironolactone 50 mg (P-score = 0.07). *S2 Fig in S1 File* shows the net split analysis. The net league table showed that Spironolactone 200 mg was associated with a significant reduction in TLC compared to Spironolactone 25 mg (SMD = -4.65, 95% CI: -5.90 to -3.40), Spironolactone 50 mg (SMD = -4.88, 95% CI: -6.13 to -3.63), TC 1% BID (SMD = -4.18, 95% CI: -5.42 to -2.94), and Tretinoin 0.05% (SMD = -4.33, 95% CI: -5.57 to -3.09), *Table 3*.

### 3.6. ILC

*Fig 3C* shows the network graph of analyzed regimens. The fixed-effect size showed that TC 1% BID significantly reduced the ILC compared to placebo (SMD = -0.27, 95% CI: -0.36 to -0.17). Other regimens showed non-significant efficacy compared to placebo, *Fig 4C*. The

**Table 1. Summary of the included studies.**

| Study ID | Study Design | Groups | Sample size, (n) | Female, N (%) | Age, years | Race, N (%) | | | | TLC, N (%) | NILC, N (%) | ILC, N (%) |
|---|---|---|---|---|---|---|---|---|---|---|---|---|
| | | | | | | White | Asian | Black | Other | | | |
| Hebert et al., 2020a | RCT phase III | TC 1% BID | 353 | 221 (62.6) | 18.0 (10–58) | 298 (84.4) | 9 (2.5) | 31 (8.8) | 15 (4.2) | 101.5 (25.12) | 59.1 (22.19) | 42.4 (11.77) |
| | | Placebo | 355 | 215 (60.6) | 18.0 (9–50) | 297 (83.7) | 10 (2.8) | 38 (10.7) | 10 (2.8) | 103.6 (26.13) | 60.7 (22.09) | 42.9 (12.31) |
| Hebert et al., 2020b | RCT phase III | TC 1% BID | 369 | 243 (65.9) | 18.0 (10–50) | 357 (96.7) | 0 (0.0%) | 7 (1.9) | 5 (1.4) | 105.7 (25.76) | 62.8 (21.37) | 42.9 (12.20) |
| | | Placebo | 363 | 221 (60.9) | 18.0 (11–42) | 348 (95.9) | 4 (1.1) | 6 (1.7) | 5 (1.4) | 104.6 (24.18) | 63.3 (20.52) | 41.3 (10.96) |
| Mazzetti et al., 2019 | RCT phase II | TC 0.1% BID | 72 | 36 (50.0) | 19 (12–43) | 58 (80.6%) | 1 (1.3%) | 12 (16.7%) | 1 (1.3%) | - | - | - |
| | | TC 0.5% BID | 76 | 42 (55.3) | 19 (12–42) | 54 (71.0%) | 3 (4.0%) | 14 (18.4%) | 2 (2.6%) | - | - | - |
| | | TC 1% BID | 70 | 38 (54.3) | 16 (12–35) | 50 (71.4%) | 4 (5.7%) | 16 (22.9%) | 0 (0.0%) | - | - | - |
| | | TC 1% ODS | 70 | 37 (52.9) | 20 (12–38) | 42 (60.0%) | 4 (5.7%) | 20 (28.6%) | 2 (2.9%) | - | - | - |
| | | Placebo | 75 | 43 (57.3) | 18 (12–35) | 53 (70.7%) | 4 (5.3%) | 12 (16.0%) | 4 (5.3%) | - | - | - |
| Trifu et al., 2011 | RCT phase III | TC 1% BID | 28 | - | 20.6±3.5 | - | - | - | - | 46.2±15 | - | 28.5 ±11.1 |
| | | Tretinoin | 30 | - | 21.2±3.4 | - | - | - | - | 48.5±17.2 | - | 29.1 ±10.4 |
| | | Placebo | 14 | - | 20.4±1.7 | - | - | - | - | 50.6±15.9 | - | 33.5 ±11.4 |
| Patiyasikunt et al., 2020 | RCT | Spironolactone 50 mg | 21 | 21 (100) | 28 (3.2) | - | - | - | - | 39.1±18.5 | 33.8 ±16.3 | 5.3±7.7 |
| | | Spironolactone 25 mg | 21 | 21 (100) | 30.9 (5.7) | - | - | - | - | 45.4±27.8 | 35.4 ±26.1 | 10±5.1 |
| | | Placebo | 21 | 21 (100) | 31.6 (5.0) | - | - | - | - | 53.2±42.5 | 43.7±41 | 9.4±5.8 |
| Muhlemman et al., 1986 | RCT | Spironolactone 200 mg | 21 | 21 (100) | - | - | - | - | - | - | - | - |
| | | Placebo | 21 | 21 (100) | - | - | - | - | - | - | - | - |
| Goodfellow et al | 1984 | Spironolactone 50 mg | 5 | - | - | - | - | - | - | - | - | - |
| | | Spironolactone 100 mg | 5 | - | - | - | - | - | - | - | - | - |
| | | Spironolactone 150 | 6 | - | - | - | - | - | - | - | - | - |
| | | Spironolactone 200 | 4 | - | - | - | - | - | - | - | - | - |
| | | Placebo | 6 | - | - | - | - | - | - | - | - | - |

TC, Topical Clascoterone; RCT, Randomized controlled trial; TLC, total acne lesion count; ILC, inflammatory lesion count; NILC, non-inflammatory lesion count; IGA, Investigator's Global Assessment

analysis showed no evidence of heterogeneity (Tau$^2$ = 0%, I$^2$ = 0%, 95% CI: 0% to 58.3%) and consistent (Q = 4.57, p = 0.95). The ranking analysis demonstrated that Tretinoin 0.05% ranked first (P-score = 0.85), followed by TC 1% BID (P-score = 0.88), Spironolactone 25 mg (P-score = 0.60), TC 1% ODS (P-score = 0.50), Placebo (P-score = 0.48), TC 0.1% BID (P-score = 0.44), TC 0.5% BID (P-score = 0.14), and Spironolactone 50 mg (P-score = 0.10). The net split analysis highlighted that TC 1% BID was associated with reduced NILC compared to TC 1% ODS (SMD = -0.25, 95% CI: -0.44 to -0.07), *S3 Fig in S1 File*. Moreover, compared

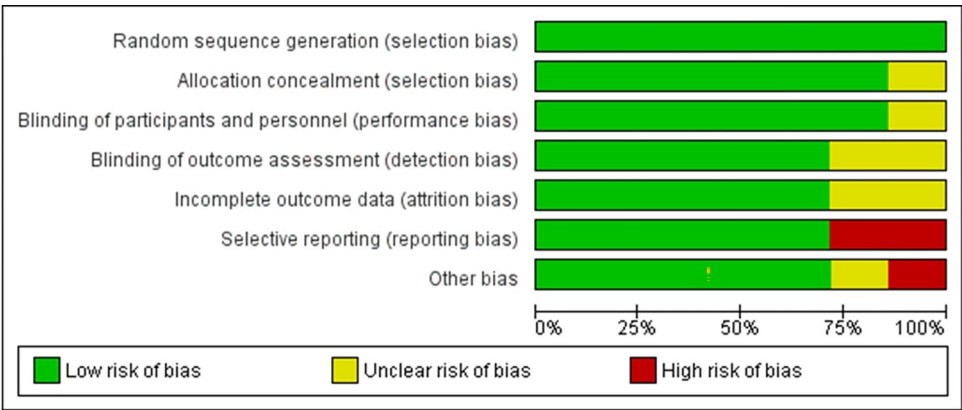

**Fig 2. Quality assessment of included studies.**

with TC 1% BID, Spironolactone 50 mg was associated with increased ILC (SMD = 0.60, 95% CI: 0.08 to 1.11), *Table 4*.

### 3.7. IGA

*Fig 3D* shows the network graph of analyzed regimens. The random-effect size showed that TC 1% BID was significantly associated with a 12-week treatment success compared to placebo

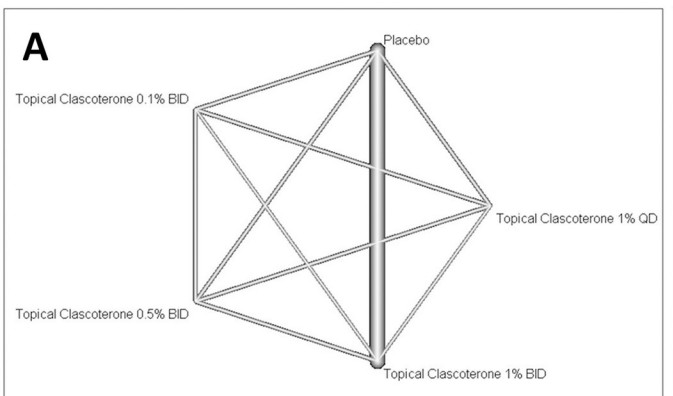

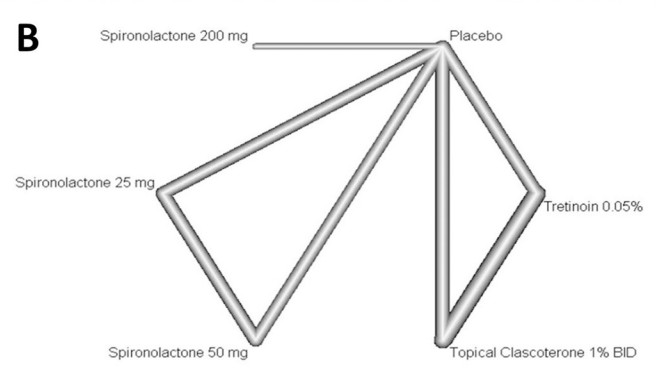

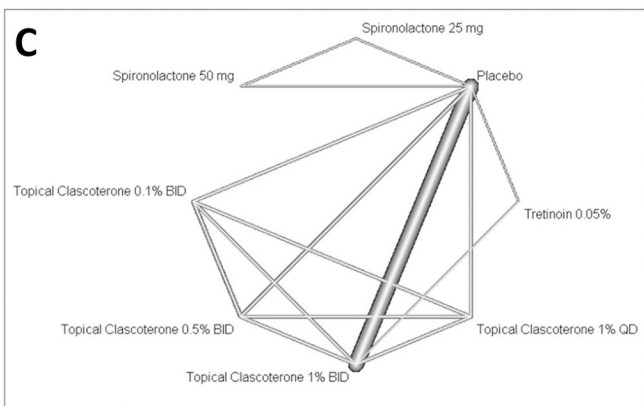

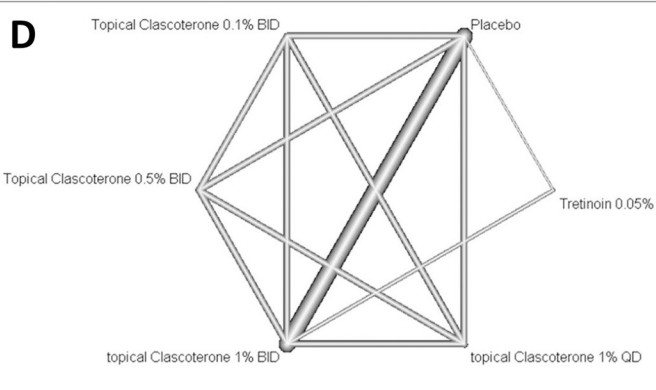

**Fig 3. Network graph.** A) NILC; B) TLC; C) ILC; D) IGA; Bold lines between studied arms means reflect the number of studies; the bolder the line the larger the number of studies.

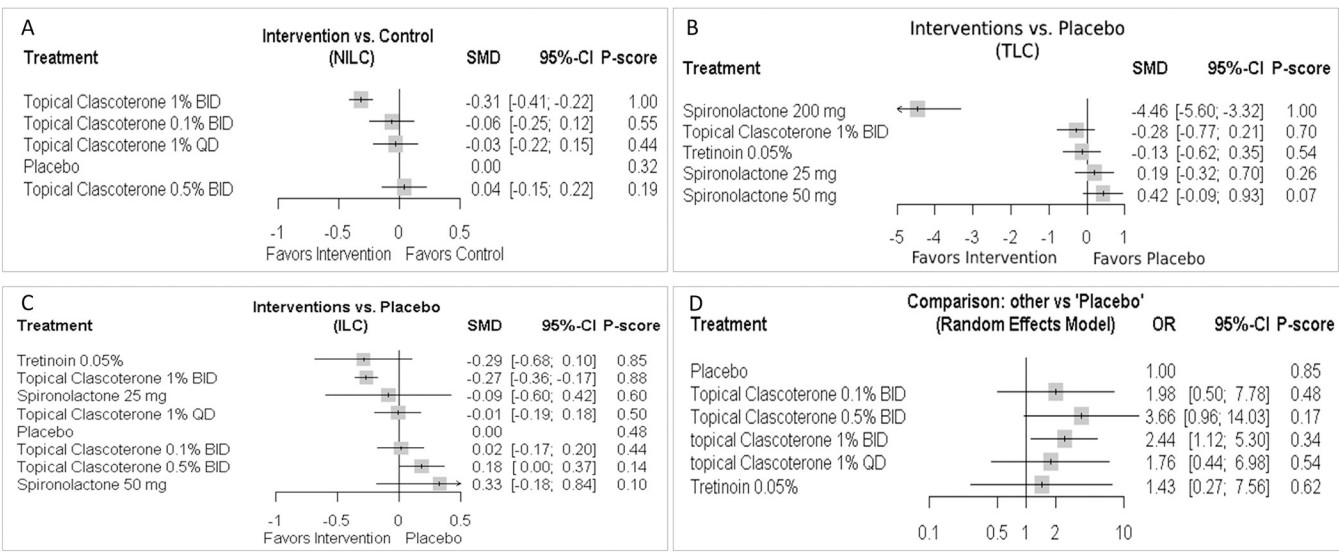

**Fig 4.** Forest Plot of A) NILC; B) TLC; C) ILC; D) IGA.

(OR = 2.44, 95% CI: 1.12 to 5.30). Other regimens showed non-significant efficacy compared to placebo, *Fig 4D*. The data showed evidence of significant heterogeneity and inconsistency (Tau$^2$ = 0.44, I$^2$ = 77.4%, 95% CI: 38.7% to 91.7%) and (Q = 13.29, p = 0.004). **S4 Fig in *S1 File*** shows the net split analysis. Moreover, compared with TC 1% BID, placebo was associated with a lower success rate (OR = 0.41, 95% CI: 0.188 to 0.89), *Table 5*. Rankogram showed that the first ranked regimen was TC 0.5% BID, followed by TC 1% BID, *Fig 5*.

### 3.8. Dose-response

In the evaluation dose response of TC, the 1% BID dosage demonstrated a significantly better outcome in reducing both NILC (*Table 2*) and ILC (*Table 4*) compared to the 0.1% BID, 0.5% BID, and 1% ODS dosages. No significant differences were noted among these lower dosages in terms of NILC and ILC reduction. With respect to the IGA of acne severity, no significant differences were observed across the various TC dosages (*Table 5*). Regarding the heterogeneity, we did not find evidence of heterogeneity in terms of the different doses in NILC and ILC outcomes (tau$^2$ = 0; tau = 0; I$^2$ = 0%; p = 0.99).

For Spironolactone, the 200 mg dosage was found to be more effective in reducing TLC compared to both the 25 mg and 50 mg dosages (*Table 3*). However, there was no notable difference in efficacy between the 25 mg and 50 mg dosages. Additionally, in terms of reducing the ILC (*Table 4*), no significant differences were identified between the 25 mg and 50 mg dosages of Spironolactone.

**Table 2. Net League table for NILC of Clascoterone at different dose.**

| Placebo | . | . | . | . |
|---|---|---|---|---|
| 0.06 (-0.12; 0.25) | TC 0.1% BID | . | . | . |
| -0.04 (-0.22; 0.15) | -0.10 (-0.31; 0.10) | TC 0.5% BID | . | . |
| **0.31 (0.22; 0.41)** | **0.25 (0.07; 0.44)** | **0.35 (0.17; 0.54)** | TC 1% BID | . |
| 0.03 (-0.15; 0.22) | -0.03 (-0.24; 0.17) | 0.07 (-0.14; 0.27) | **-0.28 (-0.47; -0.10)** | TC 1% ODS |

Data were presented as SMD and (95% CI); TC, Topical Clascoterone; NILC, non-inflammatory lesion count

**Table 3. Net league table for TLC.**

| Placebo | . | . | . | . | . |
|---|---|---|---|---|---|
| **4.46 (3.32; 5.60)** | Spironolactone 200 mg | . | . | . | . |
| -0.19 (-0.70; 0.32) | **-4.65 (-5.90; -3.40)** | Spironolactone 25 mg | . | . | . |
| -0.42 (-0.93; 0.09) | **-4.88 (-6.13; -3.63)** | -0.23 (-0.74; 0.28) | Spironolactone 50 mg | . | . |
| 0.28 (-0.21; 0.77) | **-4.18 (-5.42; -2.94)** | 0.47 (-0.23; 1.18) | **0.70 (0.00; 1.41)** | TC 1% BID | . |
| 0.13 (-0.35; 0.62) | **-4.33 (-5.57; -3.09)** | 0.32 (-0.38; 1.03) | 0.56 (-0.15; 1.26) | -0.15 (-0.58; 0.29) | Tretinoin 0.05% |

Data were presented as SMD and (95% CI); TC, Topical Clascoterone; TLC, total acne lesion count

## 4. Discussion

In this systematic review and NMA, our findings showed that TC 1% BID has potential effectiveness in reducing the NILC and ILC. Moreover, it was associated with two-times likelihood of achieving 12-week treatment success compared with a placebo. Likewise, TC 1% BID was better than TC 1% ODS, TC 0.1% BID, and TC 0.5% BID in terms of NILC reduction. On the other hand, Spironolactone 200 mg was the most effective treatment for TLC reduction. These findings highlight that TC 1% BID and Spironolactone 200 mg effectively treat patients with acne vulgaris.

Clascoterone is a novel, very effective steroidal antiandrogen that has also been shown in animal models to have localized, non-systemic effects on the skin [16]. To determine the effectiveness and safety of TC 1% cream in the treatment of mild-to-moderate acne vulgaris in comparison to placebo and topical tretinoin, Trifu et al. performed a pilot RCT [19]. It was decided to enroll just a small number of adult males and to restrict their exposure to TC to no more than eight weeks since this experiment would be the first time TC has been repeatedly administered to people. The number of participants assigned to receive placebo treatment was reduced in response to concerns raised by several ethical committees. When compared to a placebo, TC 1% significantly improved ASI, ILC, and TLC. The impact is more noticeable on the inflammatory lesions, which is interesting since it is likely attributable to the molecule's secondary anti-inflammatory action. After 2–4 weeks of therapy, the improvement in the aforementioned parameters was evident. Survival analysis also demonstrated the rapid impact of TC 1% than that of placebo and tretinoin, with a clear decrease in the number of days necessary to achieve a 50% improvement in all parameters. When compared with tretinoin, clascoterone showed greater global efficacy across all of the aforementioned metrics. Since topical retinoids have shown therapeutic success in the treatment of acne vulgaris, this observation is of special significance [22]. Patients with facial acne vulgaris had much greater reductions in

**Table 4. Net league table for ILC.**

| Placebo | . | . | . | . | . | . | . |
|---|---|---|---|---|---|---|---|
| 0.09 (-0.42; 0.60) | Spironolactone 25 mg | . | . | . | . | . | . |
| -0.33 (-0.84; 0.18) | -0.42 (-0.93; 0.09) | Spironolactone 50 mg | . | . | . | . | . |
| -0.02 (-0.20; 0.17) | -0.10 (-0.64; 0.44) | 0.31 (-0.23; 0.86) | TC 0.1% BID | . | . | . | . |
| -0.18 (-0.37; 0.00) | -0.27 (-0.81; 0.27) | 0.15 (-0.39; 0.69) | -0.17 (-0.37; 0.04) | TC 0.5% BID | . | . | . |
| **0.27 (0.17; 0.36)** | 0.18 (-0.34; 0.69) | **0.60 (0.08; 1.11)** | **0.28 (0.10; 0.47)** | **0.45 (0.26; 0.63)** | TC 1% BID | . | . |
| 0.01 (-0.18; 0.19) | -0.08 (-0.62; 0.46) | 0.34 (-0.20; 0.88) | 0.02 (-0.18; 0.23) | 0.19 (-0.02; 0.40) | **-0.26 (-0.44; -0.07)** | TC 1% ODS | . |
| 0.29 (-0.10; 0.68) | 0.20 (-0.44; 0.84) | 0.62 (-0.02; 1.26) | 0.30 (-0.12; 0.73) | **0.47 (0.04; 0.90)** | 0.02 (-0.37; 0.41) | 0.28 (-0.15; 0.71) | Tretinoin 0.05% |

Data were presented as SMD and (95% CI); TC, Topical Clascoterone; ILC, inflammatory lesion count

**Table 5. Net league table for IGA.**

| Placebo | . | . | . | . | . |
|---|---|---|---|---|---|
| 0.51 [0.13; 1.99] | TC 0.1% BID | . | . | . | . |
| 0.27 [0.07; 1.05] | 0.54 [0.12; 2.39] | TC 0.5% BID | . | . | . |
| **0.41 [0.19; 0.89]** | 0.81 [0.21; 3.22] | 1.50 [0.39; 5.80] | TC 1% BID | . | . |
| 0.57 [0.14; 2.26] | 1.13 [0.25; 5.14] | 2.08 [0.47; 9.30] | 1.39 [0.35; 5.54] | TC 1% ODS | . |
| 0.70 [0.13; 3.69] | 1.38 [0.18; 10.83] | 2.56 [0.33; 19.71] | 1.70 [0.35; 8.38] | 1.23 [0.16; 9.69] | Tretinoin 0.05% |

Data were presented as OR and (95% CI); TC, Topical Clascoterone; IGA, Investigator's Global Assessment

NILC and ILC after using TC 1% compared to those who used a vehicle cream, according to Hebert et al. study [23]. They conducted two trials (n = 1440) of patients with acne between the ages of 9 and 58 years. Treatment adherence was approximately 90% for patients applying clascoterone cream, 1%, which suggests that the treatment regimen is easy to follow and suitable for general clinical practice.

It was found by Trifu et al. that there were no statistically significant differences between treatments in terms of "success" achieved on the IGA [19]. However, compared to placebo and tretinoin, TC resulted in a significantly larger number of patients whose IGA grades were lowered from 2–3 at the screening to grades 0–1 by the end of therapy. In the study of Hebert et al., TC 1% was more effective than placebo in terms of IGA [23]. In the study of Mazzetti and his colleagues, they showed that TC 1% BID treatment had the most favorable results and was selected as the best candidate for further clinical study and development [24].

Not all patients with acne are good candidates for the currently available medications that target the androgen pathway, despite their efficacy [10, 25]. In the Hebert et al. study, the safety profile of TC 1% was comparable to that of vehicle cream, with most adverse events being low in severity [23]. In vivo investigations have shown that TC only has local antiandrogenic efficacy, suggesting the lack of systemic side effects [14, 15]. Similarly, Trifu et al. showed that there were no serious adverse events associated with TC application, no drop-outs occurred

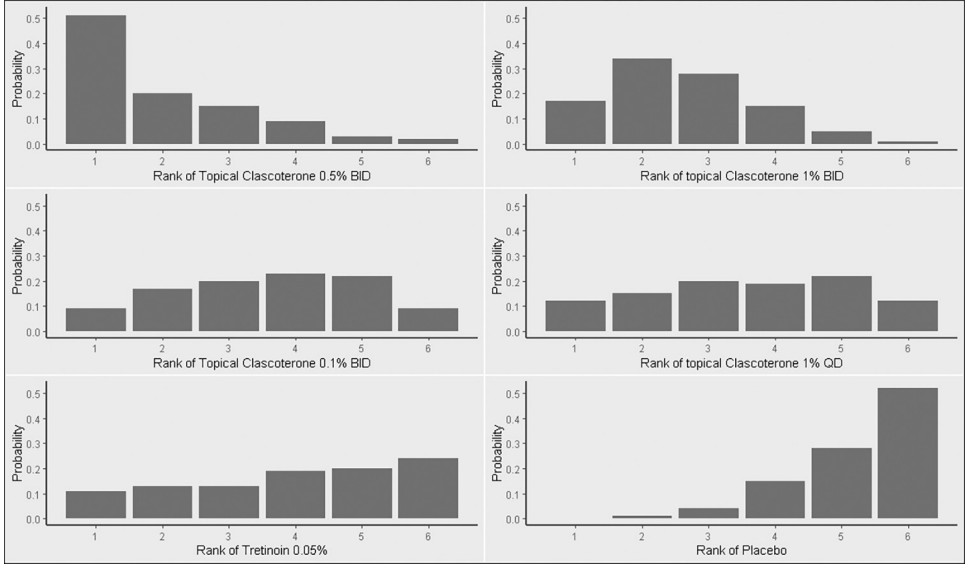

**Fig 5. Rankogram of IGA interventions.**

for safety reasons, and no differences among the groups were noticed concerning the nature and the incidence of AEs. Similarly, Mazzetti et al. showed that all TC concentrations were well-tolerated with no clinically relevant safety issues noted [24]. Two pivotal Phase 3 trials were initiated to assess the efficacy and safety of TC 1% compared with vehicles in >1400 subjects, ≥9 years of age, with moderate to severe acne (NCT 02608476) and recently concluded with final results forthcoming. An open-label extension study is underway (NCT: 02682264).

While there is evidence that antiandrogenic therapy may effectively treat moderate to severe acne in female patients [26], the treatment is not widely available due to the risks of cardiovascular disease and cancer [10]. As a potential therapy for acne in women, spironolactone has gained attention. Although the Food and Drug Administration (FDA) does not yet approve it for this specific usage, it is commonly prescribed as an alternative in the United States and elsewhere [10, 27]. On the other hand, Spironolactone is not strongly recommended by clinical guidelines since there are so few high-quality clinical studies on the topic [12]. An RCT by Patiyasikunt et al. showed considerable effectiveness in terms of decreasing TLC and improving the subjective clinical grade. Other prospective and retrospective investigations have shown outcomes consistent with these findings [12]. In comparison to other systemic acne therapies, the onset of activity for spironolactone is thought to be slower (12 weeks; range, 8–20 weeks). They also found that the benefits remained for at least a month after treatment stopped, which might be an advantage from a pharmacokinetics standpoint [28, 29]. Menstrual abnormalities were reported by 13–33% of spironolactone users, breast discomfort by 2–4%, and dizziness by 2–3%, according to a comprehensive evaluation of adverse effects [12]. Patiyasikunt et al. found that the 50-mg dose was linked to a higher likelihood of breast tenderness and dizziness but that these TRAE were mild and did not need treatment discontinuation [30].

In conclusion, TC appears to reduce both inflammatory and non-inflammatory lesion count and may lead to treatment success. Spironolactone at 200 mg showed potential effectiveness in terms of total lesion count reduction. These results suggest that both TC and Spironolactone could be beneficial in treating patients with acne vulgaris.

## Supporting information

**S1 Checklist. PRISMA 2020 checklist.**
(DOCX)

**S1 File. Figures of split analysis for NILC, TLC, ILC, and IGA.**
(PPTX)

## Acknowledgments

We would like to thank Noha Farouk Tashkandi and research platform program for their efforts in facilitating the process of this research.

## Author Contributions

**Conceptualization:** Mohammad A. Basendwh, Sarah Abdulla Bukhamsin.

**Formal analysis:** Mohammad A. Basendwh, Amer Abdulaziz Alharbi.

**Methodology:** Mohammad A. Basendwh, Amer Abdulaziz Alharbi.

**Project administration:** Sarah Abdullah Alaboud.

**Software:** Rahaf Abdulrashid Abdulwahab.

**Validation:** Rahaf Abdulrashid Abdulwahab.

**Visualization:** Rahaf Abdulrashid Abdulwahab, Sarah Abdullah Alaboud.

**Writing – original draft:** Mohammad A. Basendwh, Amer Abdulaziz Alharbi, Sarah Abdulla Bukhamsin, Rahaf Abdulrashid Abdulwahab, Sarah Abdullah Alaboud.

**Writing – review & editing:** Mohammad A. Basendwh, Amer Abdulaziz Alharbi, Sarah Abdulla Bukhamsin, Rahaf Abdulrashid Abdulwahab, Sarah Abdullah Alaboud.

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
