## [Decision Letter · Decision Letter 0]

26 Apr 2023

PONE-D-22-30388The Efficacy of Topical Clascoterone versus Systematic Spironolactone for Treatment of Acne Vulgaris: A Systematic Review and Network Meta-AnalysisPLOS ONE

Dear Dr. Basendwh,

Thank you for submitting your manuscript to PLOS ONE. After careful consideration, we feel that it has merit but does not fully meet PLOS ONE’s publication criteria as it currently stands. Therefore, we invite you to submit a revised version of the manuscript that addresses the points raised during the review process.

We look forward to receiving your revised manuscript.

Kind regards,

Kaisar Raza

Academic Editor

PLOS ONE

Journal Requirements:

  "No fund received"

Reviewers' comments:

Reviewer's Responses to Questions

**Comments to the Author**

1. Is the manuscript technically sound, and do the data support the conclusions?

Reviewer #1: Yes

Reviewer #2: Yes

Reviewer #3: Partly

2. Has the statistical analysis been performed appropriately and rigorously? 

Reviewer #1: Yes

Reviewer #2: Yes

Reviewer #3: Yes

3. Have the authors made all data underlying the findings in their manuscript fully available?

Reviewer #1: Yes

Reviewer #2: Yes

Reviewer #3: Yes

4. Is the manuscript presented in an intelligible fashion and written in standard English?

Reviewer #1: Yes

Reviewer #2: Yes

Reviewer #3: No

5. Review Comments to the Author

Reviewer #1: Reviewer(s) & Comments to Author:

The Systematic review article entitled “The Efficacy of Topical Clascoterone versus Systematic Spironolactone for Treatment of Acne Vulgaris: A Systematic Review and Network Meta-Analysis”. In this article authors have done meta-analysis computerized search through PubMed/MEDLINE, SCOPUS, and the Cochrane Library and this work sheds light on acne vulgaris treatment in clinical settings. The work seems to interesting in reading, a succinct account of the issues and concerns is as follows.

1. In introduction section authors should add the statistical estimate for cases on Acne Vulgaris per year basis.

2. What is the reason for comparing two different classes of drug based on their pharmacological effect? Spironolactone approved for used as diuretic and Clascoterone act as anti-androgen in treatment of acne. Also their route of administration is also different.

3. The title of table 2. “Direct and pooled comparisons for NILC” suggested to modify it to “Direct and pooled comparisons for NILC of Clascoterone at different dose”

4. Table 1. Lacks data of random clinical trials on male subjects.

5. Why Tretinoin comparison data for IGS is added in manuscript, when study is focused on Clascoterone and spironolactone

6. Whether the study was registered on PROSPERO or not.

7. What keywords were used and how many total search results were found for this study.

Reviewer #2: 1. Title of manuscript “The Efficacy of Topical Clascoterone versus Systematic Spironolactone forTreatment of Acne

Vulgaris: A Systematic Review and Network Meta-Analysis is a good draft.

2. In this manuscript authors are presented that topical Clascoterone and Spironolactone effectively treat patients with

acne vulgaris very interesting. Why But Spironolactone 200 mg was the most effective? Explain the details.

3. Figures1-5, its look blurry images, Check it.

4. A through grammatical check, Space, Commas (,), spelling mistakes must be performed for the entire manuscript.

Reviewer #3: The manuscript under consideration needs major revisions.

Major observations:

The abstract of the manuscript is too weak to explain about the study

In systematic reviews and meta-analysis, methodology plays a very significant role, however, in this study methodology is not clearly explained and needs improvement

The use of abbreviations is not appropriate (all over the manuscript)

The methodology section of the manuscript has been written in the bulleted format which is not the correct way of reporting

Though the systematic review and meta anlysis included very studies, author inferred the results with very strong words in results and conclusion section (e.g., most effective)

Tables and Figures were missing with the Units of measurement

In Figure 3: significance of bold lines was not explained

Results depicted in figure 2 were not discussed in the text

Statistical results are not explained in the explicit way which renders it difficult to read and understand

6. PLOS authors have the option to publish the peer review history of their article (what does this mean?). If published, this will include your full peer review and any attached files.

Reviewer #1: No

Reviewer #2: No

Reviewer #3: No

---

## [Author Response · Author response to Decision Letter 0]

6 Jul 2023

Dear Editor,

We would like to thank you and the reviewers for your efforts in reviewing our manuscript. 

We have addressed all of these valuable comments. Please find the point-to-point response: 

Reviewer Comment Response

Reviewer 1 The Systematic review article entitled “The Efficacy of Topical Clascoterone versus Systematic Spironolactone for Treatment of Acne Vulgaris: A Systematic Review and Network Meta-Analysis”. In this article authors have done meta-analysis computerized search through PubMed/MEDLINE, SCOPUS, and the Cochrane Library and this work sheds light on acne vulgaris treatment in clinical settings. The work seems to interesting in reading, a succinct account of the issues and concerns is as follows.

1. In introduction section authors should add the statistical estimate for cases on Acne Vulgaris per year basis. Thank you for your comment and feedback, we have added two sentences regarding the prevalence and incidence of Acne vulgaris globally. 

 2. What is the reason for comparing two different classes of drug based on their pharmacological effect? Spironolactone approved for used as diuretic and Clascoterone act as anti-androgen in treatment of acne. Also their route of administration is also different. Thank you for your comment. We understand the distinctions you've pointed out between Spironolactone and Clascoterone, including their pharmacological actions and routes of administration. However, our aim in this study was to assess their efficacy in treating Acne Vulgaris - a common endpoint for both drugs.

Spironolactone, while primarily a diuretic, has shown anti-androgenic properties and has been used off-label in acne treatment. Clascoterone, on the other hand, is a novel topical anti-androgen specifically approved for acne. By comparing these two medications, our study offers new insights into their relative effectiveness for acne treatment. This could provide clinicians with valuable data when deciding on the most suitable therapy for their patients. In addition, this is a systematic review and meta-analysis, so we do not compare them originally, we only included the studies that already compared them.

We appreciate your thoughtful review and hope this response addresses your concerns.

 3. The title of table 2. “Direct and pooled comparisons for NILC” suggested to modify it to “Direct and pooled comparisons for NILC of Clascoterone at different dose” Thank you for your suggestion, modified as suggested

 4. Table 1. Lacks data of random clinical trials on male subjects. We reported the percentage of females so it will be easy to calculate the percentage of males of them. 

 5. Why Tretinoin comparison data for IGS is added in manuscript, when study is focused on Clascoterone and spironolactone The study Trifu et al., 2011 included three arms Topical Clascoterone 1% BID, placebo, and Tretinoin, we cannot exclude it from the analysis as it is already reported in the study. We did not include Tretinoin as a separate study.

 6. Whether the study was registered on PROSPERO or not. Unfortunately, the study was not registered on PROSPERO; however, we carefully searched it to make sure that there were no similar protocols.

 7. What keywords were used and how many total search results were found for this study. The used terms were “Randomized clinical trial (RCT), clascoterone, spironolactone, and acne” we also employed MeSH terms during our search. 

As we reported in our results a total of 261 relevant citations were found, of them only 7 articles were included.

Reviewer 2 1. Title of manuscript “The Efficacy of Topical Clascoterone versus Systematic Spironolactone forTreatment of Acne

Vulgaris: A Systematic Review and Network Meta-Analysis is a good draft. Thank you for your feedback. We appreciate your efforts and time.

 2. In this manuscript authors are presented that topical Clascoterone and Spironolactone effectively treat patients with

acne vulgaris very interesting. Why But Spironolactone 200 mg was the most effective? Explain the details. Thank you for your comment. This is because Spironolactone 200 mg was associated with a significant reduction in total lesion count (SMD= -4.46, 95% CI: -5.60 to -3.32), making it the most effective treatment among the studied regimens.

 3. Figures1-5, its look blurry images, Check it. This is because of the conversion to PDF, we confirm that all figures were developed according to the journal style.

 4. A through grammatical check, Space, Commas (,), spelling mistakes must be performed for the entire manuscript. Thank you for your comment. We revised the manuscript and corrected all grammar errors.

Reviewer 3 The abstract of the manuscript is too weak to explain about the study Dear reviewer, we would like to thank you for your efforts in reviewing this paper. We assure you that we followed the PRISMA checklist during reporting this study, and in our abstract, we mentioned all primary outcomes of the study, and gave a short hint about our methodology and conclusion, in addition to our objectives. We also limited the number of words available for the abstract. If you could provide us with specific advice to improve our abstract this would be great. Many thanks. 

 In systematic reviews and meta-analysis, methodology plays a very significant role, however, in this study methodology is not clearly explained and needs improvement Dear reviewer, we would like to thank you for your efforts in reviewing this paper. We assure you that we followed the PRISMA checklist during reporting this study, and we covered all applicable sections of the PRISMA Checklist. If you could provide us with specific advice to improve our methods, this would be great. Many thanks.

 The use of abbreviations is not appropriate (all over the manuscript) Thank you for your comment. We have corrected it.

 The methodology section of the manuscript has been written in the bulleted format which is not the correct way of reporting Thank you for your comment. We have corrected it.

 Though the systematic review and meta anlysis included very studies, author inferred the results with very strong words in results and conclusion section (e.g., most effective) Thank you for your comment. We have reduced the voice of confidence in our results and conclusion “TC appears to reduce both inflammatory and non-inflammatory lesion count and may lead to treatment success. Spironolactone at 200 mg showed potential effectiveness in terms of total lesion count reduction. These results suggest that both TC and Spironolactone could be beneficial in treating patients with acne vulgaris.”

 Tables and Figures were missing with the Units of measurement Thank you for your comment, we have corrected it

 In Figure 3: significance of bold lines was not explained Thank you for your comment, added to the figure legend

 Results depicted in figure 2 were not discussed in the text Please refer to the results section, subsection 3.3 Quality of the included studies

 Statistical results are not explained in the explicit way which renders it difficult to read and understand Thank you for your comment. We have rephrased it to be more readable

---

## [Decision Letter · Decision Letter 1]

9 Oct 2023

PONE-D-22-30388R1The Efficacy of Topical Clascoterone versus Systematic Spironolactone for Treatment of Acne Vulgaris: A Systematic Review and Network Meta-AnalysisPLOS ONE

Dear Dr. Basendwh,

Thank you for submitting your manuscript to PLOS ONE. After careful consideration, we feel that it has merit but does not fully meet PLOS ONE’s publication criteria as it currently stands. Therefore, we invite you to submit a revised version of the manuscript that addresses the points raised during the review process.

ACADEMIC EDITOR: Please provide proper justification to the concerns raised by Reviewer 3/4.==============================

We look forward to receiving your revised manuscript.

Kind regards,

Kaisar Raza

Academic Editor

PLOS ONE

Reviewers' comments:

Reviewer's Responses to Questions

**Comments to the Author**

1. If the authors have adequately addressed your comments raised in a previous round of review and you feel that this manuscript is now acceptable for publication, you may indicate that here to bypass the “Comments to the Author” section, enter your conflict of interest statement in the “Confidential to Editor” section, and submit your "Accept" recommendation.

Reviewer #1: (No Response)

Reviewer #3: All comments have been addressed

Reviewer #4: (No Response)

2. Is the manuscript technically sound, and do the data support the conclusions?

Reviewer #1: (No Response)

Reviewer #3: Yes

Reviewer #4: No

3. Has the statistical analysis been performed appropriately and rigorously? 

Reviewer #1: (No Response)

Reviewer #3: Yes

Reviewer #4: I Don't Know

4. Have the authors made all data underlying the findings in their manuscript fully available?

Reviewer #1: (No Response)

Reviewer #3: Yes

Reviewer #4: No

5. Is the manuscript presented in an intelligible fashion and written in standard English?

Reviewer #1: (No Response)

Reviewer #3: No

Reviewer #4: Yes

6. Review Comments to the Author

Reviewer #1: (No Response)

Reviewer #3: The methodology is still not expalined in an explicit fashion; even the route of administration is not defined. Removing the bullets does not solve the purpose, it should be written in complete sentences making a paragraph.

Reviewer #4: I come to this paper as a new reviewer on the first revision.

Please do not use the term pooled analysis - meta-analysis is quite different from pooled analysis as there is only comparison within trial and then estimates are combined - a pooled analysis does not do this.

Why are both fixed (assumption-free) and random-effects methods used - the two have very different underlying approaches. Had the Q statistic been higher in the fixed case would a random-effects model have been used - in which case a random effects model was the method of analysis and the results are those for a random effects analysis with low Q.

In a network meta-analysis one issue is that one needs to know the patients are comparable and not contraindicated for one or other treatment. Otherwise in a cancer scenario one may combine fit and unfit patients and conclude erroneously that intensive treatment be given to unfit patients. Please explain how this was ensured here.

Given the doses here can one model the dose response relationship? Confidence intervals are wide here and it is incorrect to dichotomise significance as one cannot rule out meaningful differences and the 1% qd result seems out of line with the bid result. Given that qd looks like qid maybe od is a better abbreviation here for clarity.

Is information given in the publication on prior lines of therapy?

7. PLOS authors have the option to publish the peer review history of their article (what does this mean?). If published, this will include your full peer review and any attached files.

Reviewer #1: **Yes: **RAHUL SHUKLA

Reviewer #3: **Yes: **Dr Anamika

Reviewer #4: No

---

## [Author Response · Author response to Decision Letter 1]

17 Nov 2023

Dear Editor, 

We would like to thank you for your time and efforts. Please find the point-to-point response: 

Reviewer #3 

Comment: The methodology is still not expalined in an explicit fashion; even the route of administration is not defined. Removing the bullets does not solve the purpose, it should be written in complete sentences making a paragraph. 

Response: Thank you for your comment. We have added the route of administration, it is topical Clascoterone or 

Tretinoin and oral Spironolactone. The eligibility criteria were rewritten to be as one paragraph

Reviewer #4

Comment: I come to this paper as a new reviewer on the first revision. Please do not use the term pooled analysis - meta-analysis is quite different from pooled analysis as there is only comparison within trial and then estimates are combined - a pooled analysis does not do this.

Response: Thank you for your comment. We have addressed this point and removed “Pooled analysis”.

Comment: Why are both fixed (assumption-free) and random-effects methods used - the two have very different underlying approaches. Had the Q statistic been higher in the fixed case would a random-effects model have been used - in which case a random effects model was the method of analysis and the results are those for a random effects analysis with low Q.

Response: Thank you for your comment, based on the Cochrane Handbook instructions, we used the fixed-effects (plural) model by default to calculate weighted SMDs and 95% Confidence Intervals (CI) in NMA. In cases of heterogeneity among the included studies, a random-effects model was employed. The I2 and Tau2 statistics were utilized to express the inter-trial heterogeneity.

Comment: In a network meta-analysis one issue is that one needs to know the patients are comparable and not contraindicated for one or other treatment. Otherwise in a cancer scenario one may combine fit and unfit patients and conclude erroneously that intensive treatment be given to unfit patients. Please explain how this was ensured here.

Response: Dear Reviewer, we would like to thank you for highlighting this, we confirm that the baseline characteristics of all patients across the included studies did not show that any of these regimens are contraindicated in the other group.

Comment: Given the doses here can one model the dose response relationship? Confidence intervals are wide here and it is incorrect to dichotomise significance as one cannot rule out meaningful differences and the 1% qd result seems out of line with the bid result. Given that qd looks like qid maybe od is a better abbreviation here for clarity.

Response: Thank you for your comment. We have added a new section “3.8. Dose-response”, based on your suggestions

Comment: Is information given in the publication on prior lines of therapy?

Response: Yes, please refer to the third paragraph in the introduction section.

---

## [Decision Letter · Decision Letter 2]

30 Nov 2023

PONE-D-22-30388R2The Efficacy of Topical Clascoterone versus Systematic Spironolactone for Treatment of Acne Vulgaris: A Systematic Review and Network Meta-AnalysisPLOS ONE

Dear Dr. Basendwh,

Thank you for submitting your manuscript to PLOS ONE. After careful consideration, we feel that it has merit but does not fully meet PLOS ONE’s publication criteria as it currently stands. Therefore, we invite you to submit a revised version of the manuscript that addresses the points raised during the review process.Please submit your revised manuscript by Jan 14 2024 11:59PM. If you will need more time than this to complete your revisions, please reply to this message or contact the journal office at plosone@plos.org. Please include the following items when submitting your revised manuscript:A rebuttal letter that responds to each point raised by the academic editor and reviewer(s). You should upload this letter as a separate file labeled 'Response to Reviewers'.A marked-up copy of your manuscript that highlights changes made to the original version. You should upload this as a separate file labeled 'Revised Manuscript with Track Changes'.An unmarked version of your revised paper without tracked changes. You should upload this as a separate file labeled 'Manuscript'.

We look forward to receiving your revised manuscript.

Kind regards,

Kaisar Raza

Academic Editor

PLOS ONE

Reviewers' comments:

Reviewer's Responses to Questions

**Comments to the Author**

1. If the authors have adequately addressed your comments raised in a previous round of review and you feel that this manuscript is now acceptable for publication, you may indicate that here to bypass the “Comments to the Author” section, enter your conflict of interest statement in the “Confidential to Editor” section, and submit your "Accept" recommendation.

Reviewer #3: All comments have been addressed

Reviewer #4: (No Response)

2. Is the manuscript technically sound, and do the data support the conclusions?

Reviewer #3: Yes

Reviewer #4: Partly

3. Has the statistical analysis been performed appropriately and rigorously? 

Reviewer #3: Yes

Reviewer #4: I Don't Know

4. Have the authors made all data underlying the findings in their manuscript fully available?

Reviewer #3: Yes

Reviewer #4: No

5. Is the manuscript presented in an intelligible fashion and written in standard English?

Reviewer #3: Yes

Reviewer #4: Yes

6. Review Comments to the Author

Reviewer #3: (No Response)

Reviewer #4: Thank you for your response to my previous comments. I have a few remaining comments

1 The use of the word "pooled". There are still instances here, e. Section 3.4 line 4 (where the word should be excised- the data did not show heterogeneity" - the test here does not show homogeneity but rather shows no evidence of heterogeneity - these are two different concepts) - the same thing is in sections 3.5 and 3.6; Legend to Table 2, 3, 4, 5. I think in all these cases the "pooled" analysis results are the network meta-analytic estimates?

2. In terms of type of meta-analysis used then you are using random effects throughout as this reduces to fixed effects if Q<= number of degrees of freedom because of dividing by zero or negative numbers- it would make it more consistent if this was stated as otherwise it looks like you are picking and choosing.

3. This response has not made it into the paper as I can see - please add this explanation to demonstrate mathodological validity.

4. QD remains a potentially misleading abbreviation as it could be read as qid. I would still request the use of ODS for once daily

5. For dose response we need evidence of heterogeneity between doses instead of dichotomising significance. Please add this

7. PLOS authors have the option to publish the peer review history of their article (what does this mean?). If published, this will include your full peer review and any attached files.

Reviewer #3: No

Reviewer #4: No

---

## [Author Response · Author response to Decision Letter 2]

14 Jan 2024

Dear Editor, 

We would like to thank you for your time and efforts. Please find the point-to-point response:

Reviewer #3 No Response 

Response: Thank you for your feedback. 

Reviewer #4: Thank you for your response to my previous comments. I have a few remaining comments

1 The use of the word "pooled". There are still instances here, e. Section 3.4 line 4 (where the word should be excised- the data did not show heterogeneity" - the test here does not show homogeneity but rather shows no evidence of heterogeneity - these are two different concepts) - the same thing is in sections 3.5 and 3.6; Legend to Table 2, 3, 4, 5. I think in all these cases the "pooled" analysis results are the network meta-analytic estimates?

Response: Thank you for your comment. We have addressed this point and removed “Pooled analysis”. In addition, we replaced “the data did not show heterogeneity” with “shows no evidence of heterogeneity” as recommended.

2. In terms of type of meta-analysis used then you are using random effects throughout as this reduces to fixed effects if Q<= number of degrees of freedom because of dividing by zero or negative numbers- it would make it more consistent if this was stated as otherwise it looks like you are picking and choosing.

Response: Thank you for your comment, based on the Cochrane Handbook instructions, we used the fixed-effects (plural) model by default to calculate weighted SMDs and 95% Confidence Intervals (CI) in NMA. In cases of I2 is >50% and p-value is <0.10, we use random model. 

3. This response has not made it into the paper as I can see - please add this explanation to demonstrate mathodological validity.

Response: Dear Reviewer, we would like to thank you for highlighting this, we added this point to the section on characteristics of included studies.

4. QD remains a potentially misleading abbreviation as it could be read as qid. I would still request the use of ODS for once daily

Response: Thank you for your comment. We have replaced QD with ODS.

5. For dose response we need evidence of heterogeneity between doses instead of dichotomising significance. Please add this

Response: Dear Reviewer, 

We are not sure if we understand your comment correctly; however, we included the results of heterogeneity for the dose-response, if this is what you asking for. If not, please elaborate more about your request.

---

## [Decision Letter · Decision Letter 3]

22 Jan 2024

The Efficacy of Topical Clascoterone versus Systematic Spironolactone for Treatment of Acne Vulgaris: A Systematic Review and Network Meta-Analysis

PONE-D-22-30388R3

Dear Dr. Basendwh,

We’re pleased to inform you that your manuscript has been judged scientifically suitable for publication and will be formally accepted for publication once it meets all outstanding technical requirements.

Kind regards,

Kaisar Raza

Academic Editor

PLOS ONE

Additional Editor Comments (optional):

Reviewers' comments:

Reviewer's Responses to Questions

**Comments to the Author**

1. If the authors have adequately addressed your comments raised in a previous round of review and you feel that this manuscript is now acceptable for publication, you may indicate that here to bypass the “Comments to the Author” section, enter your conflict of interest statement in the “Confidential to Editor” section, and submit your "Accept" recommendation.

Reviewer #4: All comments have been addressed

2. Is the manuscript technically sound, and do the data support the conclusions?

Reviewer #4: (No Response)

3. Has the statistical analysis been performed appropriately and rigorously? 

Reviewer #4: (No Response)

4. Have the authors made all data underlying the findings in their manuscript fully available?

Reviewer #4: (No Response)

5. Is the manuscript presented in an intelligible fashion and written in standard English?

Reviewer #4: (No Response)

6. Review Comments to the Author

Reviewer #4: (No Response)

7. PLOS authors have the option to publish the peer review history of their article (what does this mean?). If published, this will include your full peer review and any attached files.

Reviewer #4: No

---

## [Editor Report · Acceptance letter]

25 Mar 2024

PONE-D-22-30388R3 

PLOS ONE

Dear Dr. Basendwh, 

I'm pleased to inform you that your manuscript has been deemed suitable for publication in PLOS ONE. Congratulations! Your manuscript is now being handed over to our production team.

Kind regards, 

on behalf of

Dr. Kaisar Raza 

Academic Editor

PLOS ONE